# Privacy-Preserving Convolutional Bi-LSTM Network for Robust Analysis of Encrypted Time-Series Medical Images

**Manjur Kolhar * and Sultan Mesfer Aldossary** 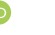

Department Computer Science, Prince Sattam Bin Abdulaziz University, Wadi Ad Dawaser 11990, Saudi Arabia; s.aldossary@psau.edu.sa
* Correspondence: m.kolhar@psau.edu.sa

**Abstract:** Deep learning (DL) algorithms can improve healthcare applications. DL has improved medical imaging diagnosis, therapy, and illness management. The use of deep learning algorithms on sensitive medical images presents privacy and data security problems. Improving medical imaging while protecting patient anonymity is difficult. Thus, privacy-preserving approaches for deep learning model training and inference are gaining popularity. These picture sequences are analyzed using state-of-the-art computer aided detection/diagnosis techniques (CAD). Algorithms that upload medical photos to servers pose privacy issues. This article presents a convolutional Bi-LSTM network to assess completely homomorphic-encrypted (HE) time-series medical images. From secret image sequences, convolutional blocks learn to extract selective spatial features and Bi-LSTM-based analytical sequence layers learn to encode time data. A weighted unit and sequence voting layer uses geographical with varying weights to boost efficiency and reduce incorrect diagnoses. Two rigid benchmarks—the CheXpert, and the BreaKHis public datasets—illustrate the framework's efficacy. The technique outperforms numerous rival methods with an accuracy above 0.99 for both datasets. These results demonstrate that the proposed outline can extract visual representations and sequential dynamics from encrypted medical picture sequences, protecting privacy while attaining good medical image analysis performance.

**Keywords:** healthcare; security; deep learning; zero watermarking; medical image

## 1. Introduction

The use of digitization has been widely adopted in the medical field due to the development of hospital standardization [1]. Digital medical pictures are produced on a daily basis by modern medical equipment [2,3]. Due to the rapid advancements in information technology, intelligent medicine and remote diagnostics are maturing [4–6]. The transmission of many medical photographs over the internet has become a standard practice [7]. X-rays, CT scans, MRI scans, and ultrasound images provide valuable information about a patient's health. These documents may also contain sensitive personal information, such as patient identifiers, which may be accessed without authorization if exposed. It is therefore crucial to develop methods to protect patient privacy without compromising the quality or utility of medical images. Using deep learning models, large amounts of data can be automatically learned to reveal complex patterns and features. As a result of this capability, they are well suited for tasks requiring privacy preservation in medical imaging. Researchers and developers have been exploring different methods for leveraging deep learning techniques in order to ensure the confidentiality and privacy of medical images. Photocopies of medical records sent over the internet are subject to theft, unauthorized use, and modification [8]. A medical picture of a patient may also contain confidential information, which may be easily leaked in this setting. Remote diagnosis and the exchange of medical images have improved with the evolution of the healthcare IT infrastructure [9]. A growing number of these methods are being used, making it increasingly important to

protect sensitive patient information, including MRI scans and other medical images, as well as electronic medical records [10,11]. Therefore, it is imperative to safeguard sensitive patient information.

During clinical examinations, time-series medical photographs demonstrate the dynamic changes in lesions. However, uploading such images to cloud servers may harm patient privacy amid growing concerns about the sharing of medical and healthcare information [12,13]. It is important to note that image scrambling encryption [14], Advanced Encryption Standard (AES) cryptosystems [15], and Rivest–Shamir–Adleman (RSA) encryption [16] only protect the data during dissemination; the cloud server must decode the data before the artificial intelligence algorithm can be applied. Due to the fact that real data can be accessed by the cloud server, these methods do not address the privacy issue. In recent research, neural networks have been used to analyze encrypted photos. As a result of their ability to compute encrypted pictures and perform well, homomorphic encryption-based privacy-preserving deep learning models are popular. In most algorithms, only individual encrypted images are calculated, making it difficult to encode discriminative time-related data. Studies of lesion dynamics are also conducted using time-series medical images. The uniqueness of medical issues and the rate of missed diagnoses should be taken into consideration when developing these approaches. Clinically, reducing the incorrect diagnosis rate is more important than improving accuracy, since missed evaluations may result in missed treatment timing, making subsequent therapy more challenging and lowering 5-year survival rates.

In order to anonymize or de-identify medical images, deep learning models are commonly used. To accomplish this, sensitive information, such as patient names, dates of birth, and other identifiable features, are removed or obfuscated while preserving the diagnostic value of the images. Using deep learning algorithms, sensitive regions can be detected and blurred or removed from images, making them suitable for research, sharing, or analysis while protecting patient privacy. Using deep learning models, it is possible to generate synthetic medical images that mimic real patient data, while ensuring the privacy of the patient. On the basis of existing medical image datasets, these models are trained to learn the underlying patterns and characteristics. Synthetic images can be used for a variety of purposes, such as algorithm development, without exposing patient information. Our work has made the following significant contributions:

1. This article proposes evaluating homomorphic-encrypted time-series medical pictures with a convolutional Bi-LSTM network. Encrypted frames have discriminative spatial characteristics extracted using convolutional blocks.
2. A weighted unit and sequence voting layer integrate geographical various weights in the suggested technique.
3. This study compares the recommended technique to a zero-watermarking solid system that meets security issues during medical photo storage and transmission, notably lesion zone protection. This comparison shows that the suggested framework protects the privacy and improves medical picture analysis.

The remainder of this article is organized into the following sections: Section 2 summarizes relevant work that examines CAD techniques for analyzing medical picture time series and numerous studies that address the privacy-preservation issue. In Section 3, we explain in depth our suggested CNN+ Bi-LSTM. The experimental design, the metrics used to evaluate it, the outcomes of the experiments, and comparisons with other recently disclosed approaches are described in Sections 4 and 5. The essay finishes with suggestions for further study in Section 6.

## 2. Related Works

Wang et al. [17] used traditional ML to diagnose breast cancer in digital mammograms using data collected at the Tumor Hospital of Liaoning Province. Two ML techniques are involved—a single-layer neural network (ELM) and a traditional support vector machine (SVM). While a DNN-based method was not used in this work, it opened the path to

employing deep learning models to carry out automated breast cancer screening in the future. The DCNN has been used on mammographic pictures by Shen et al. [18] to improve the identification of breast cancer. Resnet-50 and VGG-16 were utilized for training, while the CBIS-DDSM [19] dataset of 2478 mammography pictures was used for testing. In the ResNeSt [20], the fresh brain MR dataset was generously supplied by Ruijin Hospital, Shanghai Jiao Tong University School of Medicine, and it was used by Zhang et al. [21] to present ResNetSAt. This focus-oriented deep convolution neural network successfully detected malignancy. The CBAM's spatial-attention sub-section helped them do this.

CAD algorithms, a newly developed auxiliary diagnosis tool, might be widely used for time-series medical picture analysis. The authors [22] used a CNN with an LSTM to enhance surgical workflow identification using discriminative visual information and temporal variables. LSTM performed well in mammography image classification [23]. Reference [24] used convolutional, deconvolutional, and LSTM layers to categorize breast cancer pictures. According to the literature, LSTM and Gate Recurrent Unit (GRU) recurrent neural networks may instinctively recognize prostate cancer and myocardial infarction [25,26]. The current study uses deep learning-based CAD algorithms to interpret time-series medical photos.

Homomorphic encryption allows actions on ciphertexts deprived of decoding to evade revealing the plaintext [27]. Fully homomorphic encryption (FHE) allowed free calculations on ciphertexts for the initial time, according to Reference [28]. Over the past decade, various FHE variants have been developed to increase computation performance and privacy. The Brakerski/Fan-Vercauteren (BFV) plan [29] is the most effective fully homomorphic encryption program and encourages arbitrary multiplication and addition to encrypted messages [30]. The elegant/simple BFV approach performs well in cloud-based and secure technology [31,32].

Natsheh et al. [33] presented an efficient technique for encrypting and decrypting DICOM medical pictures using the Advanced Encryption Standard (AES). The created sequences using chaotic maps have remarkable characteristics as security keys due to their pseudo randomness, ergodicity, and beginning value responsiveness. A medical picture encryption technique based on selective chaos was presented by Kanso et al. [34]. Each iteration of this method consists of block-based shifting and masking phases. An input picture is shuffled and masked using chaotic cat maps. Using chaos theory, Song et al. [35] demonstrated a method for encrypting medical pictures securely. This approach employs a bit-level shuffling algorithm and a replacement mechanism in the permutation process to safeguard the images. Ding et al. [36] suggested a deep neural network called DeepEDN to encrypt and decode medical pictures. To secure medical images, we first use a Cycle-Generative Adversarial Net (Cycle-GAN) as the central learning system to change them from the plain arena into the target domain. The decryption process is performed via an updated network. Instead of unlocking the entire image, a region of interest (ROI)-mining network is employed to retrieve the relevant parts selectively.

Many academics have focused on using GAN-based approaches in various applications since 2014 when Goodfellow et al. [37] first presented the idea. The adversarial discriminator and generator make up the GAN network [38]. The former takes a snapshot of the data's distribution, while the latter adapts to identify anomalies in the data. Image creation [36], image segmentation [37], image super-resolution [38], and image-to-image translation are just some of the many areas where GAN-based algorithms have been shown to deliver state-of-the-art outcomes. To transform from one picture to another, Yi et al. [39] employ a conditional generative adversarial network (CGAN). It is demonstrated that this method outperforms prior art in picture synthesis using label maps, object reconstruction using edge maps, and colorization.

An epistemological framework [40] provides the foundational principles and perspectives that guide how knowledge is understood, acquired, validated, and communicated within a particular field of study or inquiry. It essentially outlines the philosophy of knowledge within that field and shapes the methods and approaches used to generate knowledge. In [41], their investigation sheds light on both the theoretical foundations and the practical

implications of ethical considerations and shared responsibility in the realm of healthcare and technology integration.

The learning network may be trained using the DualGAN [42] technique using two unlabeled pictures. DualGAN takes two sets of unlabeled pictures as input to assist many image-to-image transformation tasks and simultaneously learns two trustworthy image transformation networks. To accomplish the image transformation job using unpaired pictures, Cycle-GAN is presented in [43]. The Cycle-Gan can train two different GAN models at once. One model learns the mapping from class A to class B, while another knows the reverse. When these two mappings are combined, the loss is rethought. Adversarial loss is key to GAN's success since it ensures that produced pictures differentiate from target images. To accomplish the "Image-to-Image transformation," the negative loss is utilized to learn the mapping from the "source domain images" to the "target domain images.

## 3. Methods and Materials

Features of deep neural networks that do not leak private information are discussed here. The MORE homomorphic encryption system is the foundation of the proposed technology, which allows traditional neural network models to be trained and used directly on homomorphically secured information [44,45].

### 3.1. Problem Formulation

Let us define the problem of privacy-preserving in medical images using deep learning mathematically as follows: Given a set of sensitive medical images I = $\{I_1, I_2, \ldots, I_n\}$ with corresponding patient identifiers P = $\{P_1, P_2, \ldots, P_n\}$, where $I_i$ represents an individual image and $P_i$ represents the patient identifier associated with image $I_i$. The goal is to develop a deep learning-based framework *F* that can preserve the privacy of the medical images while maintaining their diagnostic value. The framework *F* should consist of a set of privacy-preserving techniques that can be applied to the medical images to protect sensitive patient information.

Let us denote the privacy-preserving function as $PP(I, P)$, which takes the set of medical images I and their corresponding patient identifiers P as input and outputs a transformed set of images $I' = \{I'_1, I'_2, \ldots, I'_n\}$ with preserved privacy. The transformed images $I'$ should satisfy the following conditions: The patient identifiers $P' = \{P'_1, P'_2, \ldots, P'_n\}$ associated with the transformed images $I'$ should not reveal the identity of the patients in the original set. In other words, there should be no direct link between the transformed images and their respective patient identifiers. The transformed images $I'$ should retain sufficient diagnostic information to enable effective analysis and diagnosis. The privacy-preserving techniques applied to the images should not degrade the quality or utility of the medical images.

To achieve privacy preservation in medical images using deep learning, the framework *F* should leverage the power of deep learning algorithms to develop techniques that can transform the images *I* while satisfying the anonymity and utility preservation requirements. The objective is to find an optimal privacy-preserving function $PP * (I, P)$ that maximizes the preservation of privacy while maintaining the diagnostic value of the transformed images, subject to any additional constraints or requirements specific to the application domain. Mathematically, the problem can be formulated as:

$$PP * (I, P) = arg \, max \, PP(I, P), \tag{1}$$

subject to constraints and requirements specific to privacy preservation, such as anonymity and utility preservation. The solution to the problem involves designing and training deep learning models, developing appropriate privacy-preserving techniques, and evaluating the effectiveness of the framework *F* in terms of privacy preservation and diagnostic performance using suitable evaluation metrics.

### 3.2. Dataset

The CheXpert (see Figure 1) dataset [46] is used for our investigations; it is a huge dataset with 224,316 chest X-rays from 65,240 individuals. (a) atelectasis, (b) cardiomegaly, (c) consolidation, (d) edema, and (e) pleural effusion are the five kinds that react to various thoracic diseases. There will be no effects on privacy leaks from our re-initialization of the fully connected layer and fixes to the other convolutional layers [1]. Ten thousand radio graphs are used for training and 234 are used for testing.

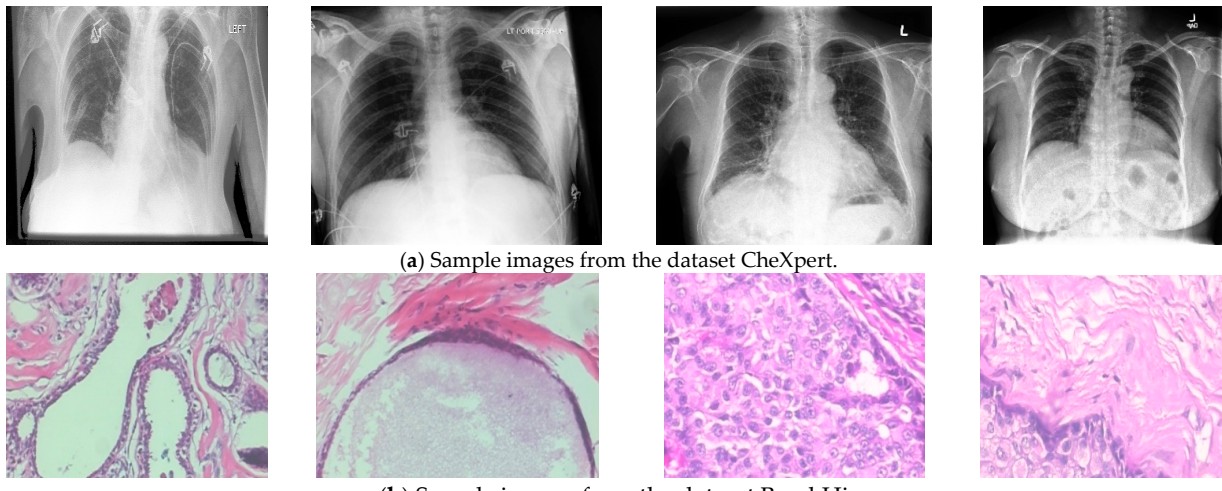

(**a**) Sample images from the dataset CheXpert.

(**b**) Sample images from the dataset BreakHis

**Figure 1.** Sample images from the dataset used in this study.

The Breast Cancer Histopathological Image Classification (BreakHis) database contains 9109 photos of breast tumor tissue, taken at $40\times$, $100\times$, $200\times$, and $400\times$ magnification levels and gathered from 82 individuals. There are now 5429 malignant samples and 2480 benign samples (all $700 \times 460$ pixels in size, 3-channel RGB, 8-bit depth, PNG format). This database was compiled in Parana, Brazil, at the P&D Laboratory of Pathological Anatomy and Cytopathology. There are two primary categories of BreaKHis tumors—benign and malignant. When a tumor lacks malignant features, such as cellular atypia, mitosis, breakdown of basement membranes, metastasis, etc., it is said to be histologically benign. Benign tumors are those that are slow-growing and are confined to one area. The invasion and destruction of neighboring structures (known as "local invasion") and metastasis to other parts of the body (known as "metastasis") are hallmarks of malignant tumors, another name for cancer.

### 3.3. Methodology

In recent years, deep learning has been used to analyze medical data with remarkable results. Despite the apparent complexity of deep learning models, they can be reduced to iterative blocks of computation based on a handful of elementary arithmetic over rational integers. The majority of state-of-the-art achievements in deep learning have been achieved using deep neural network models that employ just a small subset of possible operations. It is possible to extend the capabilities of neural network models to include ciphertext operations using the MORE scheme's homomorphic characteristic.

Figure 2 depicts the suggested process that makes use of HE and deep learning. The training data are encrypted using a private key before being processed. After that, the plaintext is separated from the processing unit and remains isolated on the side of the data source, while the ciphertext is used exclusively by the deep learning-based model. All inside network functions are structured to ensure usability on ciphertext input, and the MORE encryption method is homomorphic and allows floating-point arithmetic right away, so the system can be trained immediately on ciphertext information using the conventional training process.

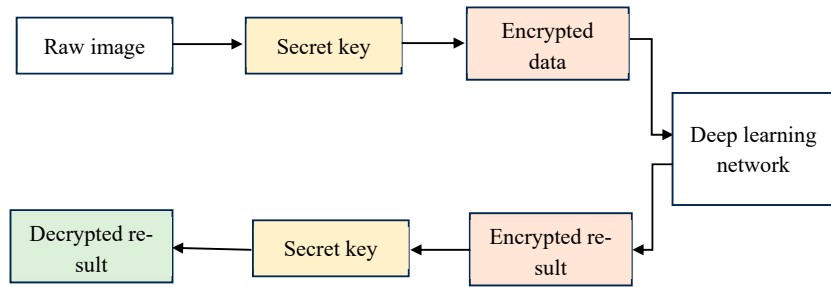

**Figure 2.** Workflow of the recommended deep learning-based application that protects the privacy and uses homomorphic encryption.

Model predictions are encrypted and can only be decoded by the owner of the secret key. After the training period has concluded, the model's encrypted form can be used to make predictions about fresh encrypted instances using the same key that was used during training. The MORE cryptosystem utilizes symmetric keys. As a result, the technique generates a secret key that can be used to encrypt plaintext data as well as decode ciphertext data as shown in the Algorithm 1.

---

**Algorithm 1 of MORE (Matrix Operation for Randomization or Encryption)**

---

**Secret Key Generation**
**Input:**
  None
**Output:**
  Secret key *SK*
**Steps:**
1. Random Matrix Generation: $R \in R^{(n \times n)}$
2. Inverse Matrix: $R_{inv} = R^{-1}$
3. Secret Key: $SK = R_{inv}$
4. Generate a random matrix R of size $(n \times n)$ with elements from a suitable key space.
5. Compute the inverse matrix $R_{inv}$ of R.
6. Set $SK = R_{inv}$.
7. Output *SK* as the secret key.

**MORE Encryption:**
**Input:**
  Plain text matrix P, Secret key *SK*
**Output:**
  Encrypted matrix C
**Steps:**
1. Plain Text Matrix: $P \in R^{(m \times n)}$
2. Encrypted Matrix: $C = P * SK$
3. Compute the matrix multiplication $C = P * SK$.
4. Output C as the encrypted matrix.

**MORE Decryption:**
**Input:**
  Encrypted matrix C, Secret key *SK*
**Output:**
  Decrypted matrix P
**Steps:**
1. Encrypted Matrix: $C \in R^{(m \times n)}$
2. Decrypted Matrix: $P = C * SK$
3. Compute the matrix multiplication $P = C * SK$.
4. Output P as the decrypted matrix.

---

### 3.4. Convolutional Bi-LSTM

The CNN has been widely used in the recognition of patterns in pictures and the detection of objects in pictures. The key benefit of CNN is its ability to automatically identify the hierarchical characteristics of incoming images. It eliminates the need for manual feature extraction, which is time-consuming and difficult. A CNN architecture is composed of three layers—convolution, pooling, and fully connected. As a result of merging the layers above, convolution blocks comprised of CLs and PLs are generated for the extraction of features from an input picture. A CNN architecture is created by linking together many convolution blocks. In the construction of a CNN for a regression or classification problem, FCLs are typically used as the final layer. Figure 3 illustrates the conventional CNN-BiLSTM design.

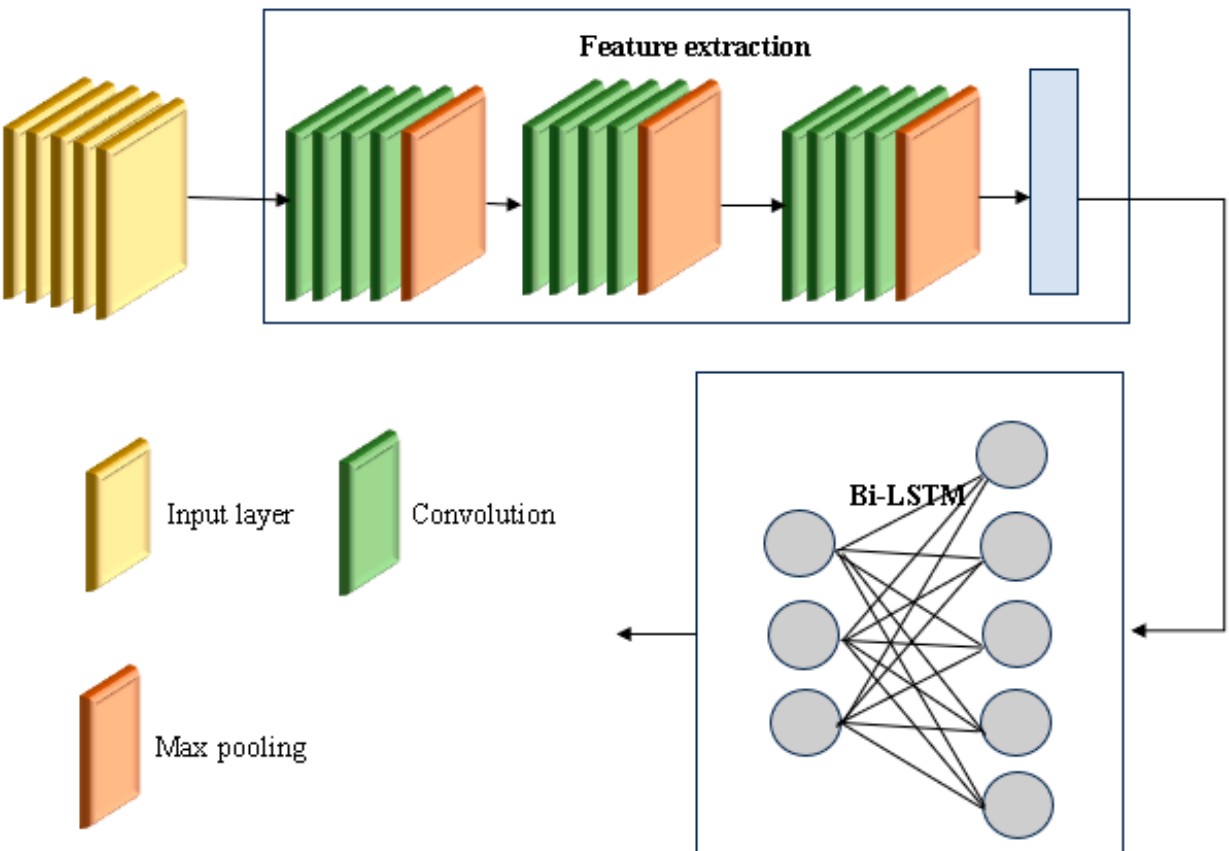

**Figure 3.** CNN + Bi-LSTM structure.

An important role is played by the convolutional layer in a CNN setup. 'Convolution kernel' is a filter series applied to the input image's or feature map's dimensions at this layer. As mentioned above, the convolution kernel is considered to be a feature extractor since it is able to extract information that is naturally present in the input picture or the output characteristic map. Convolution is a mathematical procedure in which an image is input and a kernel is output.

$$y_k = X \otimes K_k + b_k. \tag{2}$$

In the following formula, 'X' represents the input image, '$K_k$' represents the $k$th convolution kernel in CL, '$b_k$' represents the bias term, '$y_k$' represents the $k$th output feature map, and '$\otimes$' represents the convolution operation. A non-linear activation function was

then applied to the final feature map after the convolution procedure to introduce the non-linearity. The aforementioned process may be stated mathematically as:

$$S_{x,y} = a\left(\sum_{m=0}^{M-1}\sum_{n=0}^{N-1}\sum_{p=0}^{N-1} w_{p,n,m}X_{x+n,x+p,m} + b\right). \tag{3}$$

Non-linear activation function $a(\cdot)$; output feature map node at $(x, y)$ designated by $S_{x,y}$; input pixel value $x + n, x + p, m$ designating weight and bias of convolution kernel; convolution kernel size, $k \times k$. Note that the picture at $(x, y)$ at $p$th depth can be significantly affected by the size of the kernel; $w_{p,n,m}$ and $b$ represent the network's performance. Large kernel functions can generate duplicate processing and an increase in the computational complexity of a network, while tiny kernel functions can result in considerable information loss.

Following CL, PL downsamples the output feature map to make it smaller while still retaining a significant amount of spatial and uniform information. The mathematical expression for the pooling process is as follows:

$$P_{x,y,z} = L_{(m,n)\in r_{x,y}}(X_{m,n,x}). \tag{4}$$

$L(\cdot)$ represents the pooling operation; $P_{x,y,z}$ represents the updated value for the node located at coordinates $(x, y)$ in the z-th feature map; $r_{x,y}$ represents the pooling region encompassing coordinates $(x, y)$; and $X_{m,n,x}$ represents the node at coordinates $(x, y)$ inside the pooling region. There are several kinds of pooling operations. Maxpooling is the best option available. The maxpooling procedure takes a set of convolved features and chooses the one with the highest value inside the pooling window as the output feature.

FCLs are employed in both regression and classification tasks. A 1D feature vector is created from the results of CL/PL in FCL. Following a series of FCLs, the resultant layer of a classification issue is a softmax activation function. The categories are predicted using the FCL output and a probability score is calculated using the softmax activation function. Softmax activation function may be expressed mathematically as follows:

$$F = \sigma\left(h_n{}^{\circ}w^T + b\right). \tag{5}$$

Estimated class is represented by $F$, the total number of hidden neuron values is represented by ($'h_n'$), the element wise multiplication operator is ($'^{\circ}'$), the weight matrix is ($'w^T'$) between FCL and output layer and bias ($'b'$).

A variant of the long short-term memory (LSTM) technique used in recurrent neural networks (RNNs) is called Bi-LSTM. By adding bidirectional processing to the standard LSTM architecture, Bi-LSTM expands the model's capacity to account for both past and future information when generating predictions. The Bi-LSTM model may be defined mathematically as follows: the Bi-LSTM learns forward and backward hidden states, $hi^f$ and $hi^b$, from the input $I_t$ at each time step $t$. The forward LSTM units and the reverse LSTM units are responsible for calculating these latent states.

$$hi^f = LF\left(I_t, hi^f{}_{t-1}\right) \tag{6}$$

$$hi^b = LB\left(I_t, hi^b{}_{t+1}\right). \tag{7}$$

Here, $I_t$ represents the input at time step $t$, $hi^f{}_{t-1}$ is the previous hidden state for the forward LSTM unit, and $hi^b{}_{t+1}$ is the prior hidden state for the backward LSTM unit. Information about the past is stored in the forward hidden states ($h^f{}_t$), and data about the future are stored in the backward hidden states ($hi^b{}_t$). In BiLSTM, the forward normal

LSTM uses the same threshold calculation equations as a traditional LSTM, while the reverse normal LSTM uses the threshold design formulas described below.

$$in_t = \sigma(\omega_{int} \cdot [hi_{t-1}, I_t] + b_{int}) \tag{8}$$

$$fr_t = \sigma\left(\omega_{frt} \cdot [hi_{t-1}, I_t] + b_{frt}\right) \tag{9}$$

$$op_t = \sigma\left(\omega_{opt} \cdot [hi_{t-1}, I_t] + b_{opt}\right) \tag{10}$$

$$mt_t = fr_t * mt_{t-1} + in_t * tanh(\omega_{mtt} \cdot [hi_{t-1}, I_t] + b_{mtt}). \tag{11}$$

These hidden states are concatenated to obtain the final hidden state $h_t$:

$$hi_t = \left[hi^f{}_t, hi^b{}_t\right]. \tag{12}$$

The output of the Bi-LSTM model, the hidden state $hi_t$, is then utilized for prediction or other processing. Bi-LSTM's ability to handle information in both directions gives the model a head start when considering the long-term context of a prediction. This is especially beneficial in situations like traffic forecasting, when past events and anticipated ones can have a significant impact on the present. Bi-LSTM has demonstrated an enhanced performance in a number of sequence prediction applications, particularly traffic flow forecasting, by virtue of its incorporation of bidirectional processing. It may take into account historical and future data simultaneously, allowing for the identification of long-term dependencies in the data.

If you want your data-driven model to function at its best, you will need to keep a tight eye on its training phase. If the optimization is not done properly, the resulting network might not be able to accurately represent the training set or generalize to novel data. Two well-known learning-based issues that significantly impact the effectiveness of the model on a new dataset are overfitting and underfitting. Knowing when to quit exercising is crucial for avoiding these complications. Preventing the model's efficacy from deteriorating by defining early termination conditions based on the error of a validation dataset is a frequent tactic. In particular, training can be halted if the error on a held-out dataset does not decrease with time or if the difference between training and validation errors increases. The error analysis determines the halting criterion in both approaches. When working with ciphertext data, these tactics are becoming increasingly unworkable, despite being easily accepted during the training phase on plaintext data. The selected cryptosystem prevents the error metric from being utilized in a conditional statement, and the metric itself is a ciphertext.

Models that are used to ensure users' privacy are trained for a set period of time in order to get around this restriction. Since this study's overarching objective is to determine whether or not a deep neural network can successfully function on ciphertext data without any additional training, it is possible to identify an appropriate termination condition in advance. For the purpose of utility and straightforwardness, we have chosen to perform the tests and provide findings across a rather large number of epochs. We determined both the unencrypted and encrypted forms of every assignment. The neural network was taught and interpreted on plaintext data in the first play around, while ciphertext data with all trainable parameters encoded were used in the second. The training technique, hyperparameters, and startup procedure for both the plaintext and ciphertext systems were identical. Further, the same starting values were utilized for training models on both ciphertext and plaintext data. When measuring the performance of neural network algorithms using ciphertext data from the concealed testing set, every one of the assessment metrics were computed on the decoded results.

## 4. Experimental Setup

Python's Keras module and TensorFlow2 were used to implement the suggested hybrid deep neural network. The system with the Intel(R) Core (TM) i72.2 GHz CPU and the NVidia giga texel shader extreme (GTX) 1050 configuration was used to train the suggested hybrid deep neural network. With the next set of inputs, the network that was recommended was developed: learning rate = 0.0001, minibatch size = 256, and loss function = cross entropy. After every stage of the network's training execution, the loss function is optimized using the Adam optimizer. It is important to note that the network's training epoch count has been set using an early halting technique. If validation loss does not decrease by more than a threshold value (0.001) for 10 consecutive epochs, training is stopped. The assessment takes into account the epoch's weights that represent the lowest validation loss. It is worth noting that a 4-fold cross-validation approach was used to verify the network's efficacy in this endeavor.

## 5. Result and discussion

In this research, four performance metrics—correctness, exactness, specificity, and F1 score—are used to assess the efficacy of the suggested methodology. The subsequent Equations provide a mathematical expression of the aforementioned metrics. The terms '$T_{pr}$', '$T_{nr}$', '$F_{pr}$', and '$F_{nr}$' in the corresponding equations denote, accordingly, 'positive', 'negative', 'false', and 'true'.

$$Accuracy = \frac{T_{pr} + T_{nr}}{T_{pr} + T_{nr} + F_{pr} + F_{nr}} \tag{13}$$

$$Precision = \frac{T_{pr}}{T_{pr} + F_{pr}} \tag{14}$$

$$Specificity = \frac{T_{nr}}{T_{nr} + F_{pr}} \tag{15}$$

$$F1 - score = \frac{2 \times T_{pr}}{2 \times T_{pr} + F_{nr} + F_{pr}} \tag{16}$$

The suggested hybrid network's training visuals are shown in Figure 4. Figure 5 suggests that the end of the graph, when the training and validation loss are close to zero, indicates that the network has been adequately trained. It is worth noting that the suggested hybrid network takes around 95 min to train in its entirety.

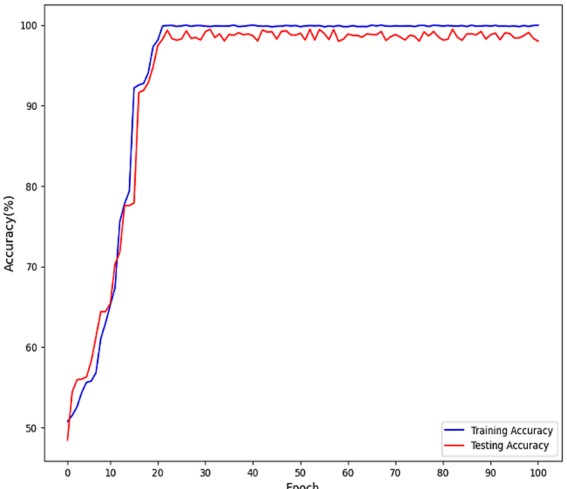
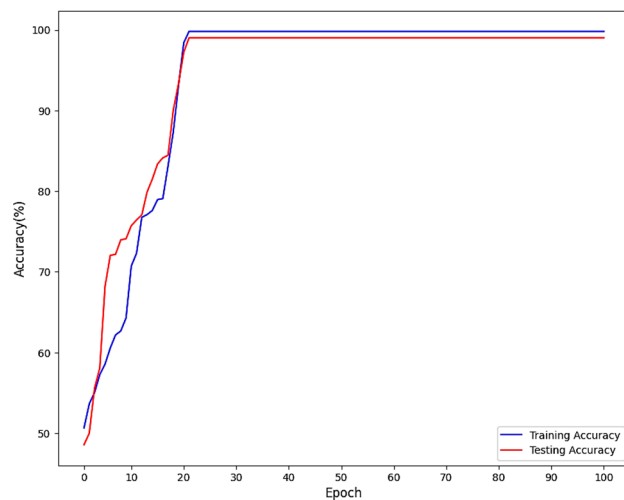

**Figure 4.** Accuracy of CNN-Bi-LSTM model on CheXpert and BreakHis datasets.

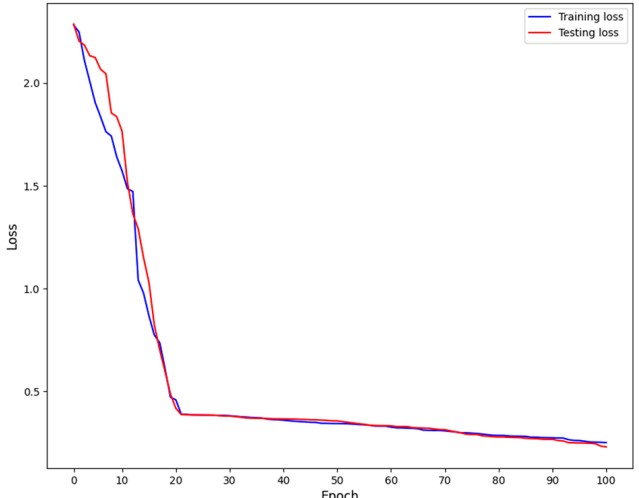 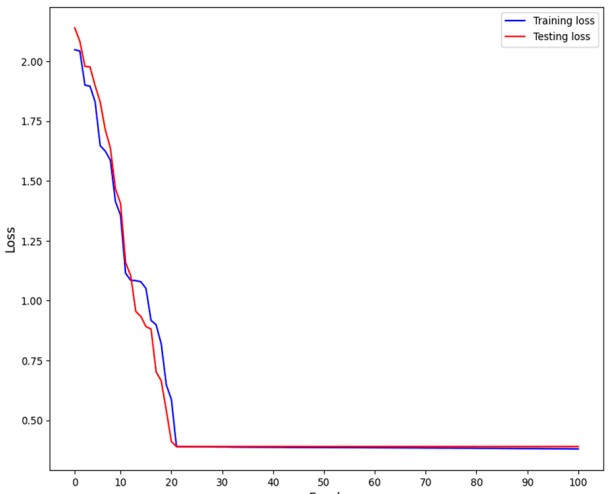

**Figure 5.** Loss of CNN-Bi-LSTM model on CheXpert and BreakHis datasets.

The loss values provided for the CNN-Bi-LSTM model on the CheXpert and BreakHis datasets are 0.39 and 0.29, respectively. Loss is a commonly used metric in machine learning that quantifies the discrepancy between the predicted output of a model and the true value. Lower loss values indicate a better agreement between the predictions and the ground truth. In Figure 5, which presumably displays the loss curve over the training epochs, we can observe that the loss starts relatively high at the beginning of training and gradually decreases as the model learns from the data. There may be fluctuations and variations in the loss during training, which is normal as the model adjusts its parameters to optimize the predictions. Overall, the loss decreases over time, indicating that the model is improving its performance on the CheXpert dataset.

The loss curve for the BreakHis dataset starts at a lower value compared to CheXpert, suggesting that the model initially performs better on this dataset. Similar to the CheXpert loss curve, there may be fluctuations and variations during training. The loss decreases consistently or stabilizes at a relatively low value, indicating that the model realizes good recital on the BreakHis dataset. We compared the proposed hybrid architecture to two existing deep architectures in terms of performance. In the first, the flattened and Bi-LSTM layers from the proposed hybrid design have been replaced with the more traditional CNN architecture. The second structure is a combination of a conventional CNN and an LSTM. The planned hybrid architecture's Bi-LSTM layers have been swapped out for regular LSTM layers in this design. The aforementioned networks have been trained, which is worth noting.

Table 1 compares the proposed hybrid (CNN-Bi-LSTM) architecture to the aforementioned deep architectures as a function of the overall amount of adjustable settings, recognition performance, and computing time. This paper shows that hybridization has resulted in a little increase in the overall number of trainable parameters in deep architecture. It is clear, however, that hybrid networks outperform regular CNNs in terms of performance. The CNN-Bi-LSTM hybrid architecture outperforms the CNN-LSTM network in terms of accuracy.

*Limitation*

The proposed approach relies on completely homomorphic encryption (HE) for the privacy-preserving analysis of medical image sequences. However, HE can be computationally expensive and may introduce additional complexity in terms of encryption and decryption operations. Deep learning algorithms, especially those involving convolutional and LSTM layers, can be computationally intensive. Performing these operations on encrypted image sequences can significantly increase the computational overhead, potentially leading to longer processing times. The framework's efficacy may vary when dealing with

heterogeneous data sources or imaging modalities, as the model may not generalize well to unseen variations. Robustness to different acquisition settings, image qualities, and imaging devices should be thoroughly investigated. While the proposed approach aims to protect patient privacy, there may still be ethical and legal concerns associated with the handling and processing of sensitive medical data, even in an encrypted form. Adherence to data protection regulations and patient consent requirements should be ensured.

**Table 1.** Result of various baseline model comparison based on performance metrics.

| Model | CheXpert | | | | BreakHis | | | |
|---|---|---|---|---|---|---|---|---|
| | Accuracy | Precision | Recall | F1-Score | Accuracy | Precision | Recall | F1-Score |
| CNN | 0.924 | 0.932 | 0.928 | 0.930 | 0.935 | 0.936 | 0.940 | 0.951 |
| LSTM | 0.944 | 0.945 | 0.952 | 0.944 | 0.945 | 0.942 | 0.948 | 0.943 |
| Bi-LSTM | 0.954 | 0.962 | 0.951 | 0.968 | 0.956 | 0.957 | 0.952 | 0.945 |
| CNN-LSTM | 0.972 | 0.984 | 0.977 | 0.976 | 0.964 | 0.962 | 0.963 | 0.970 |
| CNN-Bi-LSTM | 0.999 | 0.998 | 0.991 | 1.00 | 0.999 | 0.998 | 0.997 | 0.998 |

## 6. Conclusions

In conclusion, deep learning algorithms have shown significant potential in improving healthcare applications, particularly in the field of medical imaging diagnosis, therapy, and illness management. However, the use of sensitive medical images in deep learning models raises concerns regarding privacy and data security. Balancing the improvement of medical imaging with the protection of patient anonymity is a challenging task. Privacy-preserving approaches for deep learning model training and inference are becoming increasingly popular for addressing these concerns. State-of-the-art CAD techniques have been employed to analyze these sequential image sequences. However, the privacy issues associated with uploading medical photos to servers remain. This article presents a novel approach utilizing a convolutional Bi-LSTM network to assess completely HE time-series medical image data. The efficacy of the framework is demonstrated using two challenging benchmarks—the CheXpert dataset and the BreaKHis public dataset. The results expose that the anticipated approach outperforms numerous rival methods, achieving an impressive accuracy of above 0.99 for both datasets. This indicates that the framework successfully extracts visual depictions and captures sequential changing aspects from encrypted medical picture sequences while preserving privacy.

In addition to the proposed framework, future work should focus on further investigating and developing privacy-preserving approaches for deep learning model training and inference on sensitive medical images. Techniques such as federated learning can be explored to protect patient anonymity while maintaining the efficacy of deep learning algorithms in healthcare applications. By exploring advanced encryption methods, such as homomorphic encryption or secure multiparty computation, researchers can develop robust encryption techniques that maintain data privacy while allowing for the accurate analysis of time-series medical images. The goal is to strike a balance between maintaining privacy and preserving the integrity and usefulness of the medical data during deep learning analysis. The convergence of health policy and IoT systems presents both opportunities and challenges, particularly concerning ethical considerations and shared responsibility. Here are some actionable conclusions that the industry can consider in order to navigate these complexities while safeguarding privacy: in the future, convolutional blocks will be used for obtaining spatial characteristics from encrypted image patterns, while Bi-LSTM-based sequence evaluation layers will be used to represent temporal data. To enhance recital and reduce missed diagnoses, a weighted unit and sequence voting layer leverages geographical and temporal variables with dissimilar weights.

**Author Contributions:** Conceptualization, M.K. and S.M.A.; methodology, M.K. and S.M.A.; software, M.K. and S.M.A.; validation, M.K. and S.M.A.; formal analysis, M.K. and S.M.A.; investigation, M.K. and S.M.A.; resources, M.K. and S.M.A.; data curation, M.K. and S.M.A.; writing—original draft preparation, M.K. and S.M.A.; writing—review and editing, M.K. and S.M.A.; visualization, M.K. and S.M.A.; supervision, M.K. and S.M.A.; project administration, M.K. and S.M.A.; funding acquisition, M.K. and S.M.A. All authors have read and agreed to the published version of the manuscript.

**Funding:** This study is supported via funding from Prince Sattam bin Abdulaziz University project number (PSAU/2023/R/1444).

**Institutional Review Board Statement:** Not applicable.

**Informed Consent Statement:** Not applicable.

**Data Availability Statement:** The author proceeds within an AI approach and uses open access datasets.

**Conflicts of Interest:** The author has no conflict of interest of any form with any individual or with any organization or anybody.

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
