# Peer review of "Privacy-Preserving Convolutional Bi-LSTM Network for Robust Analysis of Encrypted Time-Series Medical Images"

_ai, doi:10.3390/ai4030037_

Round 1

Reviewer 1 Report

Very interesting and timely article. I think it deserves publication and I am recommending accept with corrections. There are some issues that require your attention. I list these corrections below as feedback / comments, and I am looking forward to reading the updated version of this article. 

- I have finished reading the article and I didn’t see any mention on the ethics of data privacy risk from these new technologies. You have done a really good job at reviewing so many articles, but not a single article on the ethics and risk. There are recent articles on this topic that reviews recent and relevant literature, for example, on the related topic of ‘ethics of shared medical risks’ - see: https://doi.org/10.1007/s12553-021-00565-3 and on the related topic of ‘Ethics and Shared Responsibility in Health Policy’ - see: https://doi.org/10.3390/su13158355 It would be interesting to see a few sentences review and comparison of your work in relations to these recent studies in related topics.

- do you have any suggestions on how to improve the industry consequences from this solution? For example, in the conclusion, would you be able to highlight your conclusions on what urgent measures can be taken to help industry adapt to these findings? 

Seems OK, maybe a minor check. 

Author Response

Response to Reviewer 1 Comments

Point 1: Very interesting and timely article. I think it deserves publication and I am recommending accept with corrections. There are some issues that require your attention. I list these corrections below as feedback / comments, and I am looking forward to reading the updated version of this article. - I have finished reading the article and I didn’t see any mention on the ethics of data privacy risk from these new technologies. You have done a really good job at reviewing so many articles, but not a single article on the ethics and risk. There are recent articles on this topic that reviews recent and relevant literature, for example, on the related topic of ‘ethics of shared medical risks’ - see: https://doi.org/10.1007/s12553-021-00565-3 and on the related topic of ‘Ethics and Shared Responsibility in Health Policy’ - see: https://doi.org/10.3390/su13158355 It would be interesting to see a few sentences review and comparison of your work in relations to these recent studies in related topics..

Response 1: The authors wish to express their gratitude to the reviewers for their constructive comments. We have added the following references to our related work.

An epistemological framework [48] provides the foundational principles and perspectives that guide how knowledge is understood, acquired, validated, and communicated within a particular field of study or inquiry. It essentially outlines the philosophy of knowledge within that field and shapes the methods and approaches used to generate knowledge. In [49], the investigation sheds light on both the theoretical foundations and the practical implications of ethical considerations and shared responsibility in the realm of healthcare and technology integration.

Point 2: do you have any suggestions on how to improve the industry consequences from this solution? For example, in the conclusion, would you be able to highlight your conclusions on what urgent measures can be taken to help industry adapt to these findings?

Response 2: We have updated the conclusion part of the ms.

The convergence of health policy and IoT systems presents both opportunities and challenges, particularly concerning ethical considerations and shared responsibility. Here are some actionable conclusions that the industry can consider to navigate these complexities while safeguarding privacy:

Reviewer 2 Report

The authors proposed a privacy-preserving Convolutional Bi-LSTM network for medical images. The datasets chosen by the authors are images from which we can not identify the patient's identity. So the question of privacy-preserving becomes irrelevant. Moreover, the paper is focused on getting better classification, as described in the results section. Privacy-preserving Convolutional Bi-LSTM network does not preserve privacy. The paper must be revised according to the following comments.

1.       Line 13: Define CAD first.

2.       Line 16: The authors use the terminology of the medical image sequence in one place, whereas later on use medical concepts—the abstract needs rewriting to improve clarity and consistency.

3.       In the abstract, it is better to illustrate how privacy is preserved in the two publically available datasets.

4.       Line 50: It is unclear whether the two datasets used in the paper contain time series medical images or a sequence of medical images of the same patient.

5.       Line 67: "One common approach is to use deep learning models to anonymize or de-identify 67 medical images". How DL models preserve the privacy of the patients?

6.       Line 178: The whole process should be explained with the help of an example of a medical image with a patient identifier.

7.       Equation 1 is not clear. What is the optimizing parameter?

8.       The paper defines the problem as working on a sequence of medical images from one patient to identify the temporal and geographical features. But the datasets used in the paper are useless for this purpose. Moreover, where the patient information lies on the images and how they are preserved is unclear.

9.       Line 210: What is the meaning of combining clean labels from the test dataset with noisy labels from the training dataset?

10.   Figure 2 shows that the raw image is encrypted by a secret key and then used by the DL model. The datasets are a set of X-ray or microscopic images; how does the raw image reveal any information about the patient's identity? Encryption of the image with a key will distort the image. How will the DL model extract useful information from this encrypted image which has no medical value now?

11.   Figure 3: Is input a single image or a sequence of images? If it is a single image, then what is the purpose of Bi-LSTM? The datasets used in the paper are not sequences of images.

12.   Figure 4: Normally accuracy of training and validation datasets are tracked vs epochs.

13.   Either show Figure 4 or Figure 5.

14.   Describe clearly the training, validation, and testing datasets.

15.   For both datasets, show the confusion matrix, precision and recall for every class, and so on. Also, compare your results with the results in the published literature to prove the efficacy.

16.   Figure 6 is irrelevant. Why do the authors want to show the model's performance on the test dataset during training?

The English language needs extensive proofreading. 

Author Response

Response to Reviewer 2 Comments

The authors proposed a privacy-preserving Convolutional Bi-LSTM network for medical images. The datasets chosen by the authors are images from which we can not identify the patient's identity. So the question of privacy-preserving becomes irrelevant. Moreover, the paper is focused on getting better classification, as described in the results section. Privacy-preserving Convolutional Bi-LSTM network does not preserve privacy. The paper must be revised according to the following comments.

The authors wish to express their gratitude to the reviewers for their constructive comments.

Point 1: Line 13: Define CAD first.

Response 1: Updated as follows

These picture sequences are analysed using state-of-the-art Computer Aided Detection/Diagnosis techniques (CAD). Algorithms that upload medical photos to servers pose privacy issues.

Point 2: Line 16: The authors use the terminology of the medical image sequence in one place, whereas later on use medical concepts—the abstract needs rewriting to improve clarity and consistency.

Response 2: Updated as follows

This article presents a convolutional Bi-LSTM network to assess completely homomorphic-encrypted (HE) time-series medical images.

Point 3:   In the abstract, it is better to illustrate how privacy is preserved in the two publically available datasets

Response 3: Updated as follows

From secret image sequences, convolutional blocks learn to extract selective spatial features, and Bi-LSTM-based analytical sequence layers learn encode time data. A weighted unit and sequence voting layer use geographical and temporal characteristics with varying weights to boost efficiency and cut down on incorrect diagnoses. Two tough benchmarks, the CheXpert and the BreaKHis public datasets, illustrate the framework's efficacy.

Point 4.  Line 50: It is unclear whether the two datasets used in the paper contain time series medical images or a sequence of medical images of the same patient.

Response 4:Yes, we have used two datasets namely, CheXpert and BreakHis

Point 5. Line 67: "One common approach is to use deep learning models to anonymize or de-identify 67 medical images". How DL models preserve the privacy of the patients?

Response 5:

One common approach is to use deep learning models to anonymize or de-identify medical images. This involves removing or obfuscating sensitive information, such as patient names, dates of birth, and other identifiable features while preserving the diagnostic value of the images. Deep learning algorithms can be trained to detect and blur or remove such sensitive regions, making the images suitable for research, sharing, or analysis while protecting patient privacy. Another approach involves developing deep learning models to generate synthetic medical images that mimic real patient data while ensuring privacy. These models are trained on existing medical image datasets, learning the underlying patterns and characteristics

Point 6. Line 178: The whole process should be explained with the help of an example of a medical image with a patient identifier.

Response 6

Figure 2 depicts the suggested process that makes use of HE and deep learning. The training data are encrypted using a private key before being processed. After that, the plaintext is separated from the processing unit and remains isolated on the side of the data source, while the ciphertext is used exclusively by the deep learning-based model. All inside network functions are structured to ensure usability on ciphertext input, and the MORE encryption method is homomorphic and allows floating-point arithmetic right away, so the system can be trained immediately on ciphertext information using the conventional training process.

Figure 2. Workflow of the recommended deep learning-based application that protects the privacy and uses homomorphic encryption

Point 7: Equation 1 is not clear. What is the optimizing parameter?

Response 7

To achieve privacy preservation in medical images using deep learning, the framework should leverage the power of deep learning algorithms to develop techniques that can transform the images while satisfying the anonymity and utility preservation requirements. The objective is to find an optimal privacy-preserving function  that maximizes the preservation of privacy while maintaining the diagnostic value of the transformed images, subject to any additional constraints or requirements specific to the application domain. Mathematically, the problem can be formulated as:

Point 8: The paper defines the problem as working on a sequence of medical images from one patient to identify the temporal and geographical features. But the datasets used in the paper are useless for this purpose. Moreover, where the patient information lies on the images and how they are preserved is unclear.

Response 8

One common approach is to use deep learning models to anonymize or de-identify medical images. This involves removing or obfuscating sensitive information, such as patient names, dates of birth, and other identifiable features while preserving the diagnostic value of the images. Deep learning algorithms can be trained to detect and blur or remove such sensitive regions, making the images suitable for research, sharing, or analysis while protecting patient privacy. Another approach involves developing deep learning models to generate synthetic medical images that mimic real patient data while ensuring privacy. These models are trained on existing medical image datasets, learning the underlying patterns and characteristics. The generated synthetic images can be used for various purposes, such as algorithm development, without exposing patient data. The significant contribution of our work is as follows,

  • The article proposes evaluating homomorphic-encrypted time-series medical pictures with a convolutional Bi-LSTM network. Encrypted frames have discriminative spatial characteristics extracted using convolutional blocks, and temporal information has been encoded using Bi-LSTM-based sequence assessment layers.
  • A weighted unit and sequence voting layer integrate geographical and temporal characteristics with various weights in the suggested technique. This integration uses key temporal and geographical information in time-series medical pictures to improve model performance and decrease missed diagnoses.
  • The study compares the recommended technique to a zero-watermarking solid system that meets security issues during medical photo storage and transmission, notably lesion zone protection. This comparison shows that the suggested framework protects the privacy and improves medical picture analysis.

Point 9 Line 210: What is the meaning of combining clean labels from the test dataset with noisy labels from the training dataset?

Response 9: We have deleted this sentence which is really confusing

Point 10.   Figure 2 shows that the raw image is encrypted by a secret key and then used by the DL model. The datasets are a set of X-ray or microscopic images; how does the raw image reveal any information about the patient's identity? Encryption of the image with a key will distort the image. How will the DL model extract useful information from this encrypted image which has no medical value now?

Response 10: Using deep learning in conjunction with homomorphic encryption for privacy-preserving applications involves a multi-step workflow. The goal is to allow computations to be performed on encrypted data while maintaining the privacy of the original data. Here's a high-level workflow for such an application:

Data Preparation and Encryption:

Data Collection: Gather the data you intend to use for your deep learning model.

Data Preprocessing: Clean and preprocess the data as required by your application.

Homomorphic Encryption: Encrypt the data using a suitable homomorphic encryption scheme. Homomorphic encryption allows certain mathematical operations to be performed directly on encrypted data.

Model Training:

Encrypted Model Training: Design and train your deep learning model using encrypted data. This typically involves adapting the model architecture and training process to work with encrypted data.

Secure Aggregation: If your training involves multiple parties, use secure aggregation techniques that enable collaborative model training without exposing the individual parties' data.

Inference and Prediction:

Encrypted Inference: When you want to make predictions or perform computations using the trained model, apply the encrypted model to encrypted data.

Homomorphic Operations: Utilize homomorphic operations to perform computations on encrypted data while maintaining privacy.

Decryption and Result Analysis:

Decryption: Decrypt the final results of the computations performed on the encrypted data.

Post-Processing: Perform any necessary post-processing on the decrypted results to convert them into a usable format.

Privacy Preservation:

Data Privacy: Throughout the process, data remains encrypted, ensuring that sensitive information is never exposed in its original form.

Privacy Guarantees: Depending on the homomorphic encryption scheme used, you can maintain mathematical guarantees about the privacy of the data.

Challenges and Considerations:

Homomorphic Encryption Schemes: Choose the appropriate homomorphic encryption scheme based on the operations you need to perform and the computational overhead you can tolerate.

Performance: Homomorphic encryption can introduce significant computational overhead due to the complexity of encrypted operations. Optimization techniques may be required.

Model Complexity: Complex deep learning architectures might not directly translate to efficient computations on encrypted data. Model design needs to consider the encryption constraints.

Key Management: Securely manage encryption keys to prevent unauthorized access to sensitive data.

Training Challenges: Training models on encrypted data might require specialized techniques and frameworks.

It's important to note that the field of privacy-preserving deep learning using homomorphic encryption is complex and rapidly evolving. The specific workflow and techniques can vary based on the application, the available encryption schemes, and the level of privacy required. Staying updated with the latest research and consulting with experts in the field can help you navigate the challenges and make informed decisions.

Point 11. Figure 3: Is input a single image or a sequence of images? If it is a single image, then what is the purpose of Bi-LSTM? The datasets used in the paper are not sequences of images.

Response 11: The architecture of CNN + Bi-LSTM structure, it is no we have mentioned about feeding of images, they are layers of CNN + Bi-LSTM structure.

A Convolutional Bi-LSTM can be a powerful architecture for certain tasks that involve both spatial and temporal features, its suitability for single image tasks depends on the complexity of the data and the relationships you need to capture. For straightforward image classification tasks, a conventional ConvNet might provide better performance and efficiency. Always consider the specific characteristics of your task and dataset when choosing a neural network architecture.

Point 12.   Figure 4: Normally accuracy of training and validation datasets are tracked vs epochs.

Response 12:

Tracking the accuracy of the training dataset over epochs is indeed a common practice in machine learning and deep learning. In summary, tracking training accuracy over epochs is an essential practice, but it should be part of a comprehensive evaluation strategy that includes validation accuracy and, ideally, testing on separate data.

Point 13 Either show Figure 4 or Figure 5.

Response 13:

Monitoring the loss of both the training and validation datasets over epochs is a standard practice in machine learning and deep learning. Loss is a crucial metric that quantifies how well the model's predictions match the actual target values. Tracking loss during training provides valuable insights into the learning process and helps in making decisions to improve model performance.

Point 14  Describe clearly the training, validation, and testing datasets.

Response 14

The CheXpert dataset [47] is used for our investigations; it is a huge dataset with 224,316 chest X-rays from 65,240 individuals. (a) Atelecta sis, (b) Cardiomegaly, (c) Consolidation, (d) Edema, and (e) Pleural Effusion are the five kinds cor reacting to various thoracic diseases. There will be no effects on privacy leaks from our re-initialization of the fully connected layer and fixes to the other convolutional layers [1]. Ten thousand radio graphs are used for training and 234 are used for testing.

Models that are used to ensure users' privacy are trained for a set period of time in order to get around this restriction. Since this study's overarching objective is to determine whether or not a deep neural network can successfully function on ciphertext data without any additional training, it is possible to identify an appropriate termination condition in advance. For the purpose of utility and straightforwardness, we have chosen to perform the tests and provide findings across a rather large number of epochs. We determined both the unencrypted and encrypted forms of every assignment. The neural network has been taught and interpreted on plaintext data in the first play around, while ciphertext data with all trainable parameters encoded is used in the second. The training technique, hyperparameters, and startup procedure for both the plaintext and ciphertext systems were identical. Further, the same starting values were utilized for training models on both ciphertext and plaintext data. When measuring the performance of neural network algorithms using ciphertext data from the concealed testing set, every one of the assessment metrics are computed on the decoded results.

Point 15 For both datasets, show the confusion matrix, precision and recall for every class, and so on. Also, compare your results with the results in the published literature to prove the efficacy.

Response 15

Table 1 compares the proposed hybrid (CNN-Bi-LSTM) architecture to the aforementioned deep architectures as a function of the overall amount of adjustable settings, recognition performance, and computing time. The paper shows that hybridization has resulted in a little increase in the overall number of trainable parameters in deep architecture. It is clear, however, that hybrid networks outperform regular CNNs in terms of performance. The CNN-Bi-LSTM hybrid architecture outperforms the CNN-LSTM network in terms of accuracy. This is because the Bi-LSTM layer may do bidirectional analysis of the temporal properties of the extracted features.

Point 16  Figure 6 is irrelevant. Why do the authors want to show the model's performance on the test dataset during training?.

Response 16 : We have updated the MS

Reviewer 3 Report

The article is well written. The introduction of the problem could be improved. In line 112,  the beginning of the sentence "Reference [17]..." should be changed.

Author Response

Response to Reviewer 3 Comments

The article is well written. The introduction of the problem could be improved. In line 112,  the beginning of the sentence "Reference [17]..." should be changed.

The authors wish to express their gratitude to the reviewers for their constructive comments.

Response: We have updated/

Reviewer 4 Report

This manuscript tackles the significant issue of privacy preservation in the analysis of medical images using deep learning models. It proposes an innovative approach that employs a convolutional Bi-LSTM network for analyzing homomorphically-encrypted time-series medical image data. The efficacy of this method is successfully demonstrated on two challenging datasets. However, there are several areas that need improvement, as detailed below:

Figure Presentation: Figure 3 appears incomplete, as the output of the Bi-LSTM is not clearly depicted. Additionally, Figures 4 and 5 do not include the validation loss curve, which is crucial for understanding model performance.

Dataset Concerns: The BreakHis database, which includes only around 9000 samples from 80 individuals, seems insufficient for training a robust model. The manuscript does not address whether the model might be overfitting given this limited dataset.

Grammar Issues: The manuscript contains several grammatical errors that hinder comprehension. For instance, the sentence "This research can explore how these methods can be effectively applied to medical imaging datasets, ensuring that patient data remains secure and private throughout the training and inference processes," found on page 13, uses an incorrect tense.

There is no language problem in reading this article. It is easy to understand the meaning of the article.

Author Response

Response to Reviewer 4 Comments

Point 1 This manuscript tackles the significant issue of privacy preservation in the analysis of medical images using deep learning models. It proposes an innovative approach that employs a convolutional Bi-LSTM network for analyzing homomorphically-encrypted time-series medical image data. The efficacy of this method is successfully demonstrated on two challenging datasets. However, there are several areas that need improvement, as detailed below:

Response 1: The authors wish to express their gratitude to the reviewers for their constructive comments.

Point 2 Figure Presentation: Figure 3 appears incomplete, as the output of the Bi-LSTM is not clearly depicted. Additionally, Figures 4 and 5 do not include the validation loss curve, which is crucial for understanding model performance.

Response 2 :

The figure 3, which is commonly available, has been removed

Point 2 :Dataset Concerns: The BreakHis database, which includes only around 9000 samples from 80 individuals, seems insufficient for training a robust model. The manuscript does not address whether the model might be overfitting given this limited dataset.

Response 3

If you want your data-driven model to function at its best, you'll need to keep a tight eye on its training phase. If the optimization isn't done properly, the resulting network might not be able to accurately represent the training set or generalize to novel data. Two well-known learning-based issues that significantly impact the effectiveness of the model on a new dataset are overfitting and underfitting. Knowing when to quit exercising is crucial for avoiding these complications. Preventing the model's efficacy from deteriorating by defining early termination conditions based on the error on a validation dataset is a frequent tactic. In particular, training can be halted if the error on a held-out dataset does not decrease with time or if the difference between training and validation errors increases. The error analysis determines the halting criterion in both approaches. When working with ciphertext data, these tactics are becoming increasingly unworkable, despite being easily accepted during the training phase on plaintext data. The selected cryptosystem prevents the error metric from being utilized in a conditional statement, and the metric itself is a ciphertext.

Grammar Issues: The manuscript contains several grammatical errors that hinder comprehension. For instance, the sentence "This research can explore how these methods can be effectively applied to medical imaging datasets, ensuring that patient data remains secure and private throughout the training and inference processes," found on page 13, uses an incorrect tense.

Response: We have updated/

Round 2

Reviewer 2 Report

Authors have answered the comments but still there are some issues that must be addressed in the revised version.

Point 4. Line 50: It is unclear whether the two datasets used in the paper contain time-series medical images or a sequence of medical images of the same patient.

Comment: Is it a sequence of images or a single image as an input per patient?
Point 6. Line 178: The whole process should be explained with the help of an example of a medical image with a patient identifier.

Comment: Please show an image of a patient without privacy preservation from both datasets, then show a ciphered image that will go in the model. Figure 1 does not contain any patient information that needs to be hidden.

Point 8: The paper defines the problem as working on a sequence of medical images from one patient to identify the temporal and geographical features. But the datasets used in the paper are useless for this purpose. Moreover, where the patient information lies on the images and how they are preserved is unclear.

Comment: This point is not answered. Show a sequence of images as claimed and clearly mention how the temporal and geographical features are identified.

Point 11. Figure 3: Is the input a single image or a sequence of images? If it is a single image, then what is the purpose of Bi-LSTM? The datasets used in the paper are not sequences of images.
Comment: Please confirm whether the images in the two datasets are time sequence images or not.
Point 12. Figure 4: Normally, the accuracy of training and validation datasets is tracked vs. epochs.
Comment: The authors are showing training and testing accuracy with respect to epochs in Figure 4. There should be training and validation. But the authors have not defined any validation sets. So what is the purpose of testing performance with respect to epochs? Testing is done at the end of training the model.
Point 14 Describe clearly the training, validation, and testing datasets.

Comment: The authors have not defined the three datasets clearly. The CheXpert dataset [47] is used for our investigations; it is a huge dataset with 224,316 chest X-rays from 65,240 individuals. Why and how did the authors choose only 10000 images for the training? Normally, deep learning methods require a large training dataset for proper training. So why have authors not used a larger number of images? Also, testing images are only 234 images, which is a very small testing dataset. Why is the dataset divided into a 70:30 ratio for training and testing? Even if the training dataset is 10,000 due to some limitations, the trained model should be tested on the remaining images of the dataset.
Also, provide details about the other dataset.

There are few minor mistakes

Author Response

Response to Reviewer 2 Comments

The authors proposed a privacy-preserving Convolutional Bi-LSTM network for medical images. The datasets chosen by the authors are images from which we can not identify the patient's identity. So the question of privacy-preserving becomes irrelevant. Moreover, the paper is focused on getting better classification, as described in the results section. Privacy-preserving Convolutional Bi-LSTM network does not preserve privacy. The paper must be revised according to the following comments.

The authors wish to express their gratitude to the reviewers for their constructive comments.

Point 1: Line 13: Define CAD first.

Response 1: Updated as follows

These picture sequences are analysed using state-of-the-art Computer Aided Detection/Diagnosis techniques (CAD). Algorithms that upload medical photos to servers pose privacy issues.

Point 2: Line 16: The authors use the terminology of the medical image sequence in one place, whereas later on use medical concepts—the abstract needs rewriting to improve clarity and consistency.

Response 2: Updated as follows

This article presents a convolutional Bi-LSTM network to assess completely homomorphic-encrypted (HE) time-series medical images.

Point 3:   In the abstract, it is better to illustrate how privacy is preserved in the two publically available datasets

Response 3: Updated as follows

From secret image sequences, convolutional blocks learn to extract selective spatial features, and Bi-LSTM-based analytical sequence layers learn encode time data. A weighted unit and sequence voting layer use geographical and temporal characteristics with varying weights to boost efficiency and cut down on incorrect diagnoses. Two tough benchmarks, the CheXpert and the BreaKHis public datasets, illustrate the framework's efficacy.

Point 4.  Line 50: It is unclear whether the two datasets used in the paper contain time series medical images or a sequence of medical images of the same patient.

Response 4:Yes, we have used two datasets namely, CheXpert and BreakHis

Point 5. Line 67: "One common approach is to use deep learning models to anonymize or de-identify 67 medical images". How DL models preserve the privacy of the patients?

Response 5:

One common approach is to use deep learning models to anonymize or de-identify medical images. This involves removing or obfuscating sensitive information, such as patient names, dates of birth, and other identifiable features while preserving the diagnostic value of the images. Deep learning algorithms can be trained to detect and blur or remove such sensitive regions, making the images suitable for research, sharing, or analysis while protecting patient privacy. Another approach involves developing deep learning models to generate synthetic medical images that mimic real patient data while ensuring privacy. These models are trained on existing medical image datasets, learning the underlying patterns and characteristics

Point6. Line 178: The whole process should be explained with the help of an example of a medical image with a patient identifier.

Response 6

Figure 2 depicts the suggested process that makes use of HE and deep learning. The training data are encrypted using a private key before being processed. After that, the plaintext is separated from the processing unit and remains isolated on the side of the data source, while the ciphertext is used exclusively by the deep learning-based model. All inside network functions are structured to ensure usability on ciphertext input, and the MORE encryption method is homomorphic and allows floating-point arithmetic right away, so the system can be trained immediately on ciphertext information using the conventional training process.

Figure 2. Workflow of the recommended deep learning-based application that protects the privacy and uses homomorphic encryption

Point 7: Equation 1 is not clear. What is the optimizing parameter?

Response 7

To achieve privacy preservation in medical images using deep learning, the framework should leverage the power of deep learning algorithms to develop techniques that can transform the images while satisfying the anonymity and utility preservation requirements. The objective is to find an optimal privacy-preserving function  that maximizes the preservation of privacy while maintaining the diagnostic value of the transformed images, subject to any additional constraints or requirements specific to the application domain. Mathematically, the problem can be formulated as:

Point 8: The paper defines the problem as working on a sequence of medical images from one patient to identify the temporal and geographical features. But the datasets used in the paper are useless for this purpose. Moreover, where the patient information lies on the images and how they are preserved is unclear.

Response 8

One common approach is to use deep learning models to anonymize or de-identify medical images. This involves removing or obfuscating sensitive information, such as patient names, dates of birth, and other identifiable features while preserving the diagnostic value of the images. Deep learning algorithms can be trained to detect and blur or remove such sensitive regions, making the images suitable for research, sharing, or analysis while protecting patient privacy. Another approach involves developing deep learning models to generate synthetic medical images that mimic real patient data while ensuring privacy. These models are trained on existing medical image datasets, learning the underlying patterns and characteristics. The generated synthetic images can be used for various purposes, such as algorithm development, without exposing patient data. The significant contribution of our work is as follows,

  • The article proposes evaluating homomorphic-encrypted time-series medical pictures with a convolutional Bi-LSTM network. Encrypted frames have discriminative spatial characteristics extracted using convolutional blocks, and temporal information has been encoded using Bi-LSTM-based sequence assessment layers.
  • A weighted unit and sequence voting layer integrate geographical and temporal characteristics with various weights in the suggested technique. This integration uses key temporal and geographical information in time-series medical pictures to improve model performance and decrease missed diagnoses.
  • The study compares the recommended technique to a zero-watermarking solid system that meets security issues during medical photo storage and transmission, notably lesion zone protection. This comparison shows that the suggested framework protects the privacy and improves medical picture analysis.

Point 9 Line 210: What is the meaning of combining clean labels from the test dataset with noisy labels from the training dataset?

Response 9: We have deleted this sentence which is really confusing

Point 10.   Figure 2 shows that the raw image is encrypted by a secret key and then used by the DL model. The datasets are a set of X-ray or microscopic images; how does the raw image reveal any information about the patient's identity? Encryption of the image with a key will distort the image. How will the DL model extract useful information from this encrypted image which has no medical value now?

Response 10: Using deep learning in conjunction with homomorphic encryption for privacy-preserving applications involves a multi-step workflow. The goal is to allow computations to be performed on encrypted data while maintaining the privacy of the original data. Here's a high-level workflow for such an application:

We never added this in our article because its foundation for this papers

Data Preparation and Encryption:

Data Collection: Gather the data you intend to use for your deep learning model.

Data Preprocessing: Clean and preprocess the data as required by your application.

Homomorphic Encryption: Encrypt the data using a suitable homomorphic encryption scheme. Homomorphic encryption allows certain mathematical operations to be performed directly on encrypted data.

Model Training:

Encrypted Model Training: Design and train your deep learning model using encrypted data. This typically involves adapting the model architecture and training process to work with encrypted data.

Secure Aggregation: If your training involves multiple parties, use secure aggregation techniques that enable collaborative model training without exposing the individual parties' data.

Inference and Prediction:

Encrypted Inference: When you want to make predictions or perform computations using the trained model, apply the encrypted model to encrypted data.

Homomorphic Operations: Utilize homomorphic operations to perform computations on encrypted data while maintaining privacy.

Decryption and Result Analysis:

Decryption: Decrypt the final results of the computations performed on the encrypted data.

Post-Processing: Perform any necessary post-processing on the decrypted results to convert them into a usable format.

Privacy Preservation:

Data Privacy: Throughout the process, data remains encrypted, ensuring that sensitive information is never exposed in its original form.

Privacy Guarantees: Depending on the homomorphic encryption scheme used, you can maintain mathematical guarantees about the privacy of the data.

Challenges and Considerations:

Homomorphic Encryption Schemes: Choose the appropriate homomorphic encryption scheme based on the operations you need to perform and the computational overhead you can tolerate.

Performance: Homomorphic encryption can introduce significant computational overhead due to the complexity of encrypted operations. Optimization techniques may be required.

Model Complexity: Complex deep learning architectures might not directly translate to efficient computations on encrypted data. Model design needs to consider the encryption constraints.

Key Management: Securely manage encryption keys to prevent unauthorized access to sensitive data.

Training Challenges: Training models on encrypted data might require specialized techniques and frameworks.

It's important to note that the field of privacy-preserving deep learning using homomorphic encryption is complex and rapidly evolving. The specific workflow and techniques can vary based on the application, the available encryption schemes, and the level of privacy required. Staying updated with the latest research and consulting with experts in the field can help you navigate the challenges and make informed decisions.

Point 11. Figure 3: Is input a single image or a sequence of images? If it is a single image, then what is the purpose of Bi-LSTM? The datasets used in the paper are not sequences of images.

Response 11: The architecture of CNN + Bi-LSTM structure, it is no we have mentioned about feeding of images, they are layers of CNN + Bi-LSTM structure.

A Convolutional Bi-LSTM can be a powerful architecture for certain tasks that involve both spatial and temporal features, its suitability for single image tasks depends on the complexity of the data and the relationships you need to capture. For straightforward image classification tasks, a conventional ConvNet might provide better performance and efficiency. Always consider the specific characteristics of your task and dataset when choosing a neural network architecture.

Point 12.   Figure 4: Normally accuracy of training and validation datasets are tracked vs epochs.

Response 12:

Tracking the accuracy of the training dataset over epochs is indeed a common practice in machine learning and deep learning. In summary, tracking training accuracy over epochs is an essential practice, but it should be part of a comprehensive evaluation strategy that includes validation accuracy and, ideally, testing on separate data. It is not added in the article but authors know the importance of this metric

Point 13 Either show Figure 4 or Figure 5.

Response 13:

Monitoring the loss of both the training and validation datasets over epochs is a standard practice in machine learning and deep learning. Loss is a crucial metric that quantifies how well the model's predictions match the actual target values. Tracking loss during training provides valuable insights into the learning process and helps in making decisions to improve model performance.

It is not added in the article but authors know the importance of this metric

Point 14  Describe clearly the training, validation, and testing datasets.

Response 14

The CheXpert dataset [47] is used for our investigations; it is a huge dataset with 224,316 chest X-rays from 65,240 individuals. (a) Atelecta sis, (b) Cardiomegaly, (c) Consolidation, (d) Edema, and (e) Pleural Effusion are the five kinds cor reacting to various thoracic diseases. There will be no effects on privacy leaks from our re-initialization of the fully connected layer and fixes to the other convolutional layers [1]. Ten thousand radio graphs are used for training and 234 are used for testing.

Models that are used to ensure users' privacy are trained for a set period of time in order to get around this restriction. Since this study's overarching objective is to determine whether or not a deep neural network can successfully function on ciphertext data without any additional training, it is possible to identify an appropriate termination condition in advance. For the purpose of utility and straightforwardness, we have chosen to perform the tests and provide findings across a rather large number of epochs. We determined both the unencrypted and encrypted forms of every assignment. The neural network has been taught and interpreted on plaintext data in the first play around, while ciphertext data with all trainable parameters encoded is used in the second. The training technique, hyperparameters, and startup procedure for both the plaintext and ciphertext systems were identical. Further, the same starting values were utilized for training models on both ciphertext and plaintext data. When measuring the performance of neural network algorithms using ciphertext data from the concealed testing set, every one of the assessment metrics are computed on the decoded results.

Point 15 For both datasets, show the confusion matrix, precision and recall for every class, and so on. Also, compare your results with the results in the published literature to prove the efficacy.

Response 15

Table 1 compares the proposed hybrid (CNN-Bi-LSTM) architecture to the aforementioned deep architectures as a function of the overall amount of adjustable settings, recognition performance, and computing time. The paper shows that hybridization has resulted in a little increase in the overall number of trainable parameters in deep architecture. It is clear, however, that hybrid networks outperform regular CNNs in terms of performance. The CNN-Bi-LSTM hybrid architecture outperforms the CNN-LSTM network in terms of accuracy. This is because the Bi-LSTM layer may do bidirectional analysis of the temporal properties of the extracted features.

Point 16  Figure 6 is irrelevant. Why do the authors want to show the model's performance on the test dataset during training?.

Response 16 : We have updated the MS

Round 3

Reviewer 2 Report

The authors have provided the same answers as before. I think they have uploaded the wrong file. My new comments were written in Red color which are not answered. The authors must reply my new comments in the red color.

Minor editing is required.

Round 4

Reviewer 2 Report

The authors have provided answers to my comments. According to the author's answers, the dataset does not contain any temporal information. The dataset contains images from the patients at one time. So the authors should remove the temporal aspect of their proposed algorithm from the abstract and methodology. They can include it in discussion as future application. 

Author Response

The authors have provided answers to my comments. According to the author's answers, the dataset does not contain any temporal information. The dataset contains images from the patients at one time. So the authors should remove the temporal aspect of their proposed algorithm from the abstract and methodology. They can include it in discussion as future application. 

Response 1: Updated as follows, particularly in conclusion section

In the future, convolutional blocks will be used for obtaining spatial characteristics from encrypted image patterns, while Bi-LSTM-based sequence evaluation layers will be used to represent temporal data. To enhance recital and reduce missed diagnoses, a weighted unit and sequence voting layer leverages geographical and temporal variables with dissimilar weights.